# The Autophagy Inhibitor Chloroquine, Alone or in Combination with mTOR Inhibitors, Displays Anti-Tumor Effects in In Vitro and In Vivo Lung Carcinoid Models

**DOI:** 10.3390/cancers13246327

**Published:** 2021-12-16

**Authors:** Adi Knigin, Shani Avniel-Polak, Gil Leibowitz, Kira Oleinikov, David J. Gross, Simona Grozinsky-Glasberg

**Affiliations:** Neuroendocrine Tumor Unit, ENETS Center of Excellence, Endocrinology and Metabolism Department, Hadassah Medical Organization and Faculty of Medicine, Hebrew University of Jerusalem, Jerusalem 91120, Israel; adi.gurarie@mail.huji.ac.il (A.K.); shani.avniel@gmail.com (S.A.-P.); GLEIB@hadassah.org.il (G.L.); kirao@hadassah.org.il (K.O.); gross@vms.huji.ac.il (D.J.G.)

**Keywords:** autophagy, mTOR inhibitors, chloroquine, lung neuroendocrine tumor

## Abstract

**Simple Summary:**

Neuroendocrine neoplasms of the lung (lung carcinoids) are often diagnosed when they are not surgically curable, and treatment options are limited. One of the approved options for treating inoperable tumors is everolimus, an mTOR inhibitor. Activation of mTOR inhibits autophagy, which is a cell survival mechanism; everolimus may paradoxically encourage cancer cell survival via stimulation of autophagy. Chloroquine, a known antimalarial compound, inhibits autophagy. Our research is focused on the hypothesis that autophagy plays a key role in the development of tumor resistance to everolimus, and that chloroquine addition to an mTOR inhibitor increases their inhibitory effect on tumor growth. In this study, we examined the effects of chloroquine alone or in combination with mTOR inhibitors on lung neuroendocrine tumor models (cell lines and mice). We have shown that chloroquine alone suppresses tumor cells’ viability and proliferation and increases their cytotoxicity and apoptosis; these effects are augmented when chloroquine is added to mTOR inhibitors. Apparently, chloroquine suppresses tumor cell growth in lung neuroendocrine neoplasms models, potentiating the effects of the mTOR inhibitors, and implying that more research is warranted to unravel its possible role in the clinical setting, in patients with advanced lung neuroendocrine neoplasms.

**Abstract:**

(1) Background: Neuroendocrine neoplasms of the lung (LNENs, lung carcinoids) are often diagnosed at an advanced stage when they are not surgically curable, and treatment options are limited. One of the approved options for treating inoperable tumors is everolimus—an mTOR inhibitor (mTORi). Activation of mTOR, among many other effects, inhibits autophagy, which is a cell survival mechanism in general, and in tumor cells in particular. Everolimus may paradoxically encourage cancer cell survival. In practice, the drug inhibits tumor development. Chloroquine (CQ) is a known antimalarial compound that inhibits autophagy. Our research is focused on the hypothesis that autophagy plays a key role in the development of tumor resistance to mTORi, and that the addition of autophagy inhibitors to mTORi exerts a synergistic effect on suppressing tumor cell proliferation. We have recently demonstrated that the combination of CQ with different mTORi increases their potency compared with mTORi alone in both in vitro and in vivo models of pancreatic NENs. In this study, we examined the effects of CQ and mTORi on in vitro and in vivo LNEN models. Aims: Testing the effects of CQ together with mTORi on cell proliferation, apoptosis, and autophagy in in vitro and in vivo LNEN models. (2) Methods: The NCI-H727 LNEN cells were treated with CQ ± mTORi. Cells’ viability and proliferation were measured using XTT and Ki-67 FACS staining. The effects of the treatments on the mTOR pathway and autophagy were examined using Western blotting. Cytotoxicity was measured using a cytotoxicity kit; apoptosis was measured by PI FACS staining and Western blotting. We further established an LNEN subcutaneous murine xenograft model and evaluated the effects of the drugs on tumor growth. (3) Results: CQ alone suppressed LNEN cells’ viability and proliferation and increased their cytotoxicity and apoptosis; these effects were augmented when CQ was added to an mTORi. We also showed the possible mechanisms for these results: on the one hand we could see a decrease in P62 levels and the absence of LC3-II (both inversely related to autophagy) following treatment with the mTORi, and on the other hand we could demonstrate an increase in their levels when CQ was added. The effect was less apparent in the murine xenograft model. (4) Conclusions: By inhibiting autophagy and inducing apoptosis, CQ suppresses tumor cell growth in LNENs. CQ potentiates mTORi effects, implying that further studies are needed in order to elucidate its possible role in tumor inhibition in patients with LNENs.

## 1. Introduction

Lung neuroendocrine neoplasms (LNENs, lung carcinoids) are rare tumors with an age-adjusted incidence rate ranging from 0.2 to 2/100,000 population/year [1], with an increasing prevalence over the past 30 years. LNENs include a large spectrum of tumors with respect to neuroendocrine morphology and differentiation, the main distinction being between the well-differentiated (typical carcinoids (TCs); atypical carcinoids (ACs)) and the poorly differentiated (small-cell or large-cell carcinoma) neoplasms [2]. Excision of potentially resectable tumors is preferred, with curative intent and better survival; complete TC resection carries 5- and 10-year survival rates of 90% and 80%, respectively, with a recurrence rate of 3–5%, whereas ACs have 5- and 10-year survival rates of 70% and 50%, respectively, with a 25% recurrence rate [3]. About 47% of LNENs are diagnosed at an advanced stage and, hence, considered inoperable [4]. Treatment options for unresectable LNENs are of limited efficacy (mainly tumor stabilization for restricted periods of time), and include somatostatin analogs and/or peptide receptor radionuclide therapy (PRRT) (for somatostatin receptor expressing tumors), targeted therapies (e.g., tyrosine kinase inhibitors, mammalian (mechanistic) target of rapamycin inhibitors (mTORi)) and, rarely, chemotherapy [5].

NENs, including LNENs, have been shown to have alterations in signal transduction pathways, such as the PI3K/Akt/mTOR pathway, which may promote tumor growth, along with increased tissue invasiveness and angiogenesis [6,7]. Known therapeutic agents inhibiting this pathway are the mTORi, rapamycin and its analogues [8,9,10]. Importantly, mTOR exists in two different complexes: mTORC1, which is the target for rapamycin and its analogues, and mTORC2, which activates AKT, thereby promoting feedback tumor cell proliferation and the development of resistance to mTORC1 inhibitors (mTORC1i) [8,9,10]. Therefore, a major drawback of the use of mTORC1 is their lack of mTORC2 inhibition, being unable to abrogate the S6K-mediated negative feedback loop downstream of mTORC1, with rebound AKT activation and autophagy-associated cell survival, leading to drug resistance [11,12]. Torin1, a global mTOR inhibitor, can overcome this undesired effect, being shown to inhibit cell proliferation more effectively [11].

Autophagy is a homeostatic, catabolic, “self-consuming” cellular process, activated in response to stresses such as starvation or hypoxia; it begins with the formation of double-membrane vesicles, known as autophagosomes, which engulf cytoplasmic components (cellular proteins and organelles). Then, the autophagosomes fuse with lysosomes, where the sequestered contents undergo degradation and recycling to sustain cellular activities [11,13]. Interestingly, it was observed that specific inhibition of mTORC1 releases the suppression of autophagosome and autolysosome formation, resulting in an increase in the number and size of autolysosomes [11,14]. Chloroquine (CQ) is an antimalarial agent that inhibits autophagy through inhibition of lysosomal hydrolase activity [15]. Lysosomal hydrolase is vital for autophagosome and lysosome fusion, without which the autophagy process is stalled; it is optimally active in an acidic environment, and CQ inhibits it by increasing the lysosomal pH. Our group has previously shown that treatment with CQ alone or in combination with an mTORi inhibited cells’ proliferation and induced apoptosis [11] in both in vitro and in vivo models of pancreatic NEN, inducing significant decreases in tumor volume in a BON1-xenograft murine model [15]. Herein, we aimed to study the effect of CQ—alone, or together with mTORi—in in vitro and in vivo LNEN models.

## 2. Materials and Methods

### 2.1. Reagents

The reagents used and their preparation were as follows: RAD001 (LC laboratories, USA) was dissolved in DMSO, then diluted with PBS to a stock solution of 10 mM and stored at −20 °C. Torin1 (Tocris, Bristol, UK) was dissolved in DMSO to yield a stock solution of 1 mM. Chloroquine (Sigma-Aldrich Ltd., Rehovot, Israel) was diluted in PBS to a concentration of 10 mM. The controls were treated with the same medium as that diluting the drugs.

### 2.2. Cell Culture

The human LNEN (well-differentiated carcinoid) cell line NCI-H727 (CRL# 5815 ATCC, Manassas, VA, USA) was cultured in RPMI-1640 medium supplemented with 10% FCS, 100 U/mL penicillin, and 100 μg/mL streptomycin (Biological Industries, Beit HaEmek, Israel). Cells were incubated in 10 cm plates at 37 °C under a 5% CO_2_ atmosphere and passaged every 3–4 days.

### 2.3. Cell Viability

The cells were seeded and grown for 24 h in 96-well plates (at a density of 2 × 10^4^ cells/per well). Then, the cells were treated with CQ (100 µM), RAD001 (100 nM), Torin1 (250 nM), and combinations thereof (CQ and RAD001, or CQ and Torin1). The concentrations of CQ and Torin1 were based on the IC_50_ at 48 h. The IC_50_ for RAD001 in NCI-H727 cells could not be determined because RAD001 did not show a significant inhibitory effect on the cells in the preliminary IC_50_ experiments. Therefore, we decided to use the RAD001 IC_50_ from our previous experience with BON-1 cells [11]. DMSO was used as a treatment for the control group.

Cells’ viability was estimated using the XTT cell proliferation assay (Biological Industries, Beit HaEmek, Israel) at 24, 48, and 72 h of treatment, using Multiskan GO, plate reader, and SkanIt RE 5.0 software (Thermo Scientific, Waltham, MA, USA). All assays were performed in 6 replicates and were repeated at least 3 times.

### 2.4. Flow Cytometry

NCI-H727 cells were seeded in 12-well tissue culture plates at a density of 5 × 10^5^ cells/well and incubated for 24 h. Then, the cells were treated with RAD001 (100 nM), Torin1 (250 nM), or CQ (100 µM) for 48 h. For proliferation analysis, the cells were washed and stained with murine anti-Ki-67 Alexa Fluor 647 antibody (BD Biosciences, New Jersey, NJ, USA) according to the manufacturer’s instructions. For apoptosis analysis, the cells were stained for Annexin V-APC and PI (propidium iodide) according to the manufacturer’s instructions (BioLegend, San Diego, CA, USA). Cells were analyzed using an LSRII flow cytometer (Becton Dickinson Bioscience, Franklin Lakes, NJ, USA). The results were processed with FCS Express 4 software (De Novo Software, Glendale, CA, USA).

### 2.5. Drug Cytotoxicity Assay

NCI-H727 cells (2 × 10^4^ cells per well) were seeded in a 96-well plate. After 24 h of incubation, they were treated with CQ (100 µM), RAD001 (100 nM), Torin1 (250 nM), and their combinations for up to 72 h. Cell toxicity was assessed at three timepoints (24, 48, and 72 h after treatment initiation) with the CellTox^TM^ Green Cytotoxicity Assay (Promega, Madison, WI, USA) on the same plate.

### 2.6. Western Blotting

NCI-H727 cells (at a density of 1 × 10^6^ cells/well) were seeded in a 6-well plate with a poor medium (2% FCS) for 24 h. On the following day, cells were treated with CQ (100 µM), RAD001 (100 nM), Torin1 (250 nM), and their combinations. After a period of 24 h, RIPA buffer with protease inhibitors and phosphatase inhibitors was added to each well (Sigma-Aldrich Ltd., Rehovot, Israel). Protein yield was quantified using the Bradford protein assay (BIO-RAD, Hercules, CA, USA). Protein samples were denatured in SDS sample buffer and separated by electrophoresis on SDS-PAGE gel, and were then transferred to a PVDF membrane. The membranes were incubated overnight with primary antibodies. The antibodies that we used included anti-phospho-p70S6K, anti-total p70S6K, anti-GAPDH, anti-LC3-II, and anti SQSTM1/P62 (Cell Signaling Technology, Danvers, MA, USA). Antibodies were prepared in Tris-buffered saline/Tween 20 (TBST) with 5% BSA and sodium azide (Sigma-Aldrich Ltd., Rehovot, Israel). After 3 washes in TBST, the membranes were incubated with a secondary antibody (peroxidase-conjugated goat anti-rabbit IgG; Jackson ImmunoResearch laboratories INC, West Grove, PA, USA) in TBST with 5% skimmed milk for 1 h at room temperature. Then, the blots were developed with the EZ-ECL detection system (Biological Industries, Beit HaEmek, Israel). For the quantification of the immunoblots, we used the ChemiDoc^TM^ Touch Imaging System and Image Lab 6.0 software (Bio-Rad, Hercules, CA, USA).

### 2.7. Subcutaneous Murine Xenograft Model

LNEN xenografts were established 5 days following the subcutaneous injection of 4 × 10^6^ NCI-H727 cells into the backs of athymic nude mice (FOXN1NU NU/NU mice, ENVIGO, Israel). Once the neoplasm size reached 130 mm^3^, the mice were randomized into 4 groups and treated for the next 13 days with (1) vehicle (PBS, 100 μL), (2) CQ (60 mg/kg), (3) RAD001 (3 mg/kg), or (4) CQ and RAD001 (60 mg/kg and 3 mg/kg, respectively). All treatments were administered via IP (intraperitoneal) injections. Tumor size was measured daily using a caliper, and its volume was calculated using the following equation: length × (width^2^)/2. We chose the drug concentrations after conducting calibration experiments with the recommended concentrations in this model, based on reviewing the literature and choosing those concentrations that suggested efficacy [15]. The animals were treated daily (except for weekends). At the completion of the experiment, the mice were euthanatized, and the tumors were excised and weighed, followed by histopathological analysis including H&E staining and validation of the neuroendocrine tumor morphology. The subcutaneous xenograft model is presented in Figure 7 (adapted from our previous publication [15]).

### 2.8. Statistical Analyses

Statistical analysis and significance of differences between groups were assessed using Student’s *t*-test. The results are shown as the mean ± SEM. Significance was taken at *p* < 0.05.

## 3. Results

### 3.1. CQ and the mTORi RAD001 and Torin1 Inhibit LNEN Cell Viability

To evaluate the effects of the drugs on cell viability, we treated NCI-H727 cells with CQ (100 µM), RAD001 (100 nM), Torin1 (250 nM), and their combinations (CQ and RAD001, or CQ and Torin1) for 24, 48, and 72 h (Figure 1).

Interestingly, at all timepoints, treatment with RAD alone did not affect cell viability, and the results were similar to those of the control group (cells with no treatment). Torin1 minimally decreased cell viability at 24 h; however, the effect increased at 48 h and stabilized at 72 h, with decreases in cell viability of ~40% and 35%, respectively. It should be noted that the decrease in cell viability by CQ was significant at all timepoints, either when used alone (by ~30%, ~50%, and ~72% at 24 h, 48 h, and 72 h, respectively), or when combined with either RAD001 (by ~35%, ~60%, and ~80%, at 24 h, 48 h, and 72 h, respectively) or with Torin1 (by ~55%, ~89%, and ~98%, at 24 h, 48 h, and 72 h, respectively). In summary, CQ—alone, or combined with mTORi for different lengths of time—decreases cell viability in a synergistic manner. As XTT is a crude measurement of the metabolic state of the cell, which indirectly reflects cell viability, we next studied the effects of the drugs on cell proliferation and apoptosis.

### 3.2. CQ, RAD001, and Torin1 Inhibit LNEN Cell Proliferation

NCI-727 cells were treated for 48 h with CQ (100 µM), RAD001 (100 nM), Torin1 (250 nM), and their combinations (CQ and RAD001, or CQ and Torin1) (Figure 2). Cell proliferation was measured by flow cytometry and Ki-67 staining. As expected, untreated cells had the highest levels of proliferation. Cell proliferation decreased by 16%, 49%, and 49% following RAD001, CQ, and Torin1 treatments, respectively. The addition of RAD001 to CQ did not show a further significant decrease in cell proliferation (46% vs. 49% with CQ alone). However, the addition of Torin1 to CQ reduced cell proliferation by 70% compared to 49% with CQ alone (Figure 2A,B). When comparing the combinations of CQ and RAD001 or CQ and Torin1 to either RAD001 or Torin1 alone, there was a significant decrease in proliferation. Altogether, these results suggest an important inhibitory effect of CQ on LNEN cell proliferation; moreover, CQ apparently potentiates the inhibitory effects of each of the mTORi used in this study.

### 3.3. CQ, Alone or in Combination with mTORi, Induces NCI-H727 Cell Death 

Following the promising inhibitory effects seen on cell viability and proliferation, we next analyzed the drugs’ effects on cell death using a cytotoxicity assay. For this purpose, we treated the cells with CQ, RAD001, and Torin1 alone, as well as with their combinations for different lengths of time, and using the aforementioned dosages (Figure 3).

Interestingly, the effect was similar for all treatment arms at 24 h. In contrast, at 48 h, RAD and Torin1 showed a mild decrease in cytotoxicity, whereas CQ—alone or in combination with RAD001 or Torin1—showed an increase in the cytotoxic effect; this pattern of increased cytotoxicity persisted at 72 h, however, showing a slight decrease in the CQ + RAD001 arm when compared with the 48 h effect. Collectively, these results indicate that CQ alone induces cell death; moreover, its addition increased the cytotoxic effects of RAD001, and especially of Torin1, with a peak effect at 48 h.

### 3.4. mTORi and CQ Have Synergistic Effects, and Their Combination Suppresses mTOR Activation

The PI3K/Akt/mTOR pathway is an important regulator of cell proliferation. The phosphorylation of P70-S6K at Thr389 (p-P70-S6K), downstream of mTORC1, is a marker of proliferation activation via mTORC1; conversely, the inhibition of mTORC1 decreases the P70-S6K phosphorylation at its Thr389 site, decreasing proliferation. Moreover, it is well established that mTORC1 is a key inhibitor of autophagy, and its inhibition induces autophagy, which is eventually blocked by the addition of CQ [11].

We next studied the effects of CQ and mTORi—alone, or in combination—on mTORC1 activity by measuring the levels of Thr389 phosphorylation in NCI-727 cells after 24 h of treatment (Figure 4; the original WB figures, including densitometry readings/intensity ratio of each band, are provided in the Appendix A).

As expected, both the mTORC1i RAD001 and the total mTORi Torin1 decreased the levels of p-P70S6K, whereas CQ alone increased the p-P70S6K phosphorylation. Importantly, the addition of RAD001 or Torin1 to CQ abolished the effect of the latter and inhibited p-P70S6K phosphorylation, synergistically suppressing mTOR activation. These results suggest that although CQ may induce mTOR activation via inhibition of autophagy, its co-administration with mTORi not only abolishes this effect, but even increases the mTORi-related mTOR pathway suppression.

### 3.5. CQ and mTORi Induce Apoptosis in NCI-H727 Cells

Inhibition of autophagy by CQ may lead to apoptosis; therefore, we studied the effects of CQ alone or together with mTORi on apoptosis in the LNEN cells via Annexin V staining followed by flow cytometry analysis. Early apoptotic cells are Annexin V-positive and PI-negative (Annexin V+/PI-), whereas late (end-stage) apoptotic cells are Annexin V/PI-double-positive. We treated NCI-H727 cells with CQ, RAD001, Torin1, and their combinations for 48 h (at the dosages mentioned in the Materials and Methods section), and looked at their effects on different stages of apoptosis (early, late, and total) (Figure 5). At 48 h, CQ increased the number of apoptotic cells by 45%, whereas the effects of Torin1 and RAD001 were milder (increasing the number of apoptotic cells by only 17% and 3%, respectively) (Figure 5A–D). The combination of CQ and RAD001 significantly increased the numbers of both early apoptotic cells (by 48%, *p* = 0.007) and total apoptotic cells (by 51%, *p* = 0.002) vs. RAD001 alone. The combination of CQ and Torin1 increased the number of apoptotic cells by 45%, inducing higher levels of late apoptotic cells (11%), but this was not statistically significant (Figure 5E, F). These results suggest that CQ induces apoptosis and potentiates the pro-apoptotic effects of the mTORi in NCI-H727 cells.

### 3.6. CQ Inhibits Autophagy and Diminishes mTORi Induction of Autophagy in NCI-H727 Cells

To estimate the status of the autophagy process, we further examined the effects of CQ and mTORi—alone and in combination—on autophagy, via Western blotting of two canonical autophagy markers: p62 and LC3-II (Figure 6; the original WB figures, including densitometry readings/intensity ratio of each band, are provided in the Appendix A). p62 is selectively degraded by autophagy, and is a widely used marker for the activation of autophagy [15]. There is a good inverse correlation between the amount of p62 and the activation of autophagy. A complementary method is to measure the autophagy flux, which is important because p62 levels may be low for unrelated reasons. The flux is estimated by measuring the levels of the autophagosome-associated protein LC3-II before and after treatment with an autophagy inhibitor such as CQ [16,17]. Upon stimulation of autophagy, cytosolic LC3 (LC3-I) is recruited to the autophagosome membrane and undergoes lipidation, resulting in a mobility shift of LC3—so-called LC3-II; therefore, the expression of LC3-II correlates with the number of autophagosomes [16]. LC3-II undergoes degradation in autolysosomes; hence, treatment with lysosomal enzyme inhibitors is expected to increase LC3-II levels [13,18,19,20,21,22,23].

Treatment with CQ increased the levels of p62, whereas RAD001 and Torin1 mildly decreased them, compared to untreated starved cells; the addition of CQ to either RAD001 or Torin1 increased p62 levels, supporting effective suppression of autophagy (Figure 6A).

Furthermore, in untreated NCI-H727 cells, LC3-II was absent, which may reflect a failure to generate autophagosomes or rapid autophagosome turnover. As expected, CQ increased LC3-II levels, indicating that autophagosome generation was intact. The addition of CQ to mTORi (both RAD001 and Torin1) induced accumulation of LC3-II compared with the mTORi alone, suggesting that CQ abolishes the mTORi-induced autophagy to some extent in this model (Figure 6B).

Altogether, our findings suggest that in NCI-H727 cells the autophagic flux is rapid, with high autophagosome turnover; moreover, they imply that CQ inhibits not only the lysosomal degradation, but also the mTORi-related activation of autophagy, finally decreasing cells’ survival and proliferation.

### 3.7. The Addition of CQ to mTORi Decreases Tumor Volume Compared to mTORi Alone in the NCI-H727 Subcutaneous Murine Xenograft Model

Next, we evaluated the drugs’ effects on tumor development using an NCI-H727 subcutaneous murine xenograft model (Figure 7). Mice treated with PBS (controls) developed large neoplasms, reaching an average size of 300 mm^3^ at the end of the study. Mice treated with CQ alone did not show a decreased neoplasm size when compared to the control group (average size 390 mm^3^), indicating that CQ alone has a limited inhibitory effect on neoplasm growth in this setting. The RAD001-treated group developed significantly smaller neoplasms than the control mice (average size 187 mm^3^). It should be noted that the addition of CQ to RAD001 induced a reduction in neoplasm size comparable to that of RAD001 alone, reaching an average size of 165 mm^3^ at the completion of the experiment (each neoplasm measurement was performed as related to its initial size). We monitored neoplasm volumes in the RAD001 and the RAD001 + CQ groups for an extended period in order to test whether the effect on neoplasm size persisted. The inhibitory effects of RAD001 and RAD001 + CQ persisted for up to ~22 days, when the tumors stopped responding to the treatment and started to progress rapidly. However, the differences between the RAD and RAD001 + CQ groups were not statistically significant. Altogether, although the effects of CQ were less impressive in the murine model than in the NCI-H727 cell line model, its addition to mTORi facilitated the decrease in tumor volume when compared with CQ alone (Figure 8).

## 4. Discussion

Since many NENs fail to persistently respond to currently available therapies, we aimed in this study to unravel the impacts of autophagy inhibition on cell proliferation and tumor progression in LNENs. Considering that autophagy may function as a stress-activated pro-survival mechanism in LNENs, we hypothesized that inhibition of autophagy using lysosomal inhibitors such as CQ may promote cell death, enhancing the anti-tumorigenic effects of mTORi and overcoming the mTORi-related drug resistance frequently acquired in clinical practice.

In the present study, we show that treatment with CQ—alone, or together with mTORi—induces a robust inhibitory effect on LNEN cell proliferation and survival in the in vitro NCI-H727 LNEN model; interestingly, while the effect of CQ alone was less impressive in the in vivo NCI-H727 murine xenograft LNEN model, it became more evident when CQ was added to RAD001 compared with RAD001 alone. These findings reveal unexplored avenues in the heterogeneity of NENs and in their response to these compounds, as well as in the behavior of refractory LNENs.

We showed initially that LNEN cell viability was reduced under treatment with CQ and Torin1, but with no significant effect following administration of RAD001; however, the inhibitory effect became powerful again when CQ was added to RAD001, and even more so when added to Torin1, compared with the effects of each of the drugs alone. These results are in concordance with our previous studies in both in vitro and in vivo pancreatic neuroendocrine neoplasm (pNEN) models, in which CQ demonstrated a strong effect; the treatment of these models with a combination of CQ with either Torin1 or NVP-BEZ235 (BEZ) decreased pNEN cells’ viability and proliferation compared with each drug alone, and increased cells’ apoptosis [11,24]. Regarding the pro-apoptotic effect in LNEN cells, it was mild for RAD001 and Torin1; it should be noted that CQ induced a significant apoptotic effect, and also potentiated the pro-apoptotic effects of the mTORi in NCI-H727 cells in the present study.

Both mTORi—Torin1 and RAD001—alone, and in combination with CQ, markedly decreased the levels of p-P70S6K, implying suppression of LNEN cell proliferation. CQ alone increased the p-P70S6K levels, consistent with the upregulation of P70S6K by CQ that has been previously reported for other tumor models, being most likely explained by the activation of alternative survival pathways involving AKT in cells in which the process of autophagy was suppressed [11]. The fact that p-P70S6K remains more suppressed when CQ is added to either RAD001 or Torin1 is interesting, as is the evidence that RAD001 alone demonstrated a smaller decrease in p-P70S6K than when combined with CQ.

A major finding of our current research is that CQ, as a single treatment, inhibits LNEN cell proliferation and promotes apoptosis. We assumed that the reason for this could be autophagy inhibition. The effect of autophagy inhibition in LNEN cells indeed appears to be significant, and apparently occurs regardless of the mTOR inhibition. Since mTOR inhibition amplifies autophagy—a pro-survival mechanism [24]—we hypothesized that adding autophagy inhibitors could enhance the anti-tumorigenic effect of the PI3K/Akt/mTOR pathway inhibitors in NEN tumor cells. By combining mTOR inhibitors with CQ—a lysosomal autophagy inhibitor—we were able to show a synergistic effect in reducing cell viability and enhancing cytotoxicity and apoptosis in LNEN cells.

The role of autophagy inhibition in LNEN cells was initially examined by Hong et al. [24], who demonstrated that inhibiting autophagy with CQ is cytotoxic for these cells, compared with a “cytostatic effect only” in non-LNEN cells. They suggested that autophagy may play a distinct role in LNEN cell survival, and that these cells might be more sensitive to autophagy inhibition than non-LNEN cells. Autophagic activity is measured biochemically by the amounts of LC3-II and p62 that accumulate in the absence or presence of lysosomal activity [24]. CQ treatment increased p62 levels, supporting effective suppression of autophagy in the LNEN cell model. Theoretically, mTORi increase autophagy, but we were unable to show a greater autophagic flux in cells treated with mTORi alone compared to the control group, possibly because the starvation that the cells experienced had already maximized their autophagy, limiting the effects of the mTORi. However, the enhanced effects obtained when mTORi were combined with CQ indicate that autophagy remains an essential process in LNEN cells, similar to the effect we have seen in the pNEN models. It should be noted that the stronger effects observed in the LNEN cells may suggest that they are more sensitive to autophagy inhibition than other NEN models. One hypothesis could be that starvation by itself is a strong enough trigger for the induction of autophagy, and the additive effect of mTORi is minor. Although more studies are warranted in order to determine whether the favorable effects seen in this model system can be attributed to increased autophagy, our results demonstrate that mTORi can induce autophagy in vitro and in vivo in NEN models, and suggest that these effects should be further examined under autophagy-enriched conditions.

Another significant finding of the present in vitro study is the powerful inhibitory effect on LNEN cells by CQ—alone, or when combined with mTORi—in agreement with our previous results in CQ-treated pNEN models. Interestingly, the effect was different in the in vivo LNEN model, when mice harboring LNEN xenografts treated with CQ alone did not show a decrease in the neoplasm size, but did develop smaller tumors when CQ was added to RAD001. Although the reason for this discrepancy remains unclear, we believe that it is possibly related to physiological differences between cell lines and xenografts that impact drug absorption, distribution, metabolism, and excretion, as well as drug–drug interaction. Moreover, we [11,15] and others [25] have shown that CQ can be added to RAD001, inhibiting tumor cell viability in parallel with increasing apoptosis and reducing autophagy.

In summary, the present study suggests that inhibition of autophagy by CQ robustly inhibits LNEN cell survival and proliferation, potentiating the anti-tumor effects of mTORi via stimulation of apoptosis. Since CQ was reported to have pleiotropic effects on cells [26,27,28,29,30], our results further support the hypothesis that in the LNEN models, cell apoptosis is related to the inhibitory effect of CQ on autophagy, and not to an alternative effect of CQ on the cells. This may have important clinical implications for considering the use of CQ as a therapeutic strategy aiming to overcome mTORi-related drug resistance, and represents an important consideration for the development of appropriate clinical trials in patients with metastatic or refractory LNENs. Further studies are needed in order to understand the precise role of CQ in the therapeutic arsenal of patients with metastatic lung NENs.

The main limitation of our study is that it was conducted using a single LNEN cell line (of typical carcinoids) for both the in vitro and in vivo studies (as we repeatedly failed in growing an additional atypical carcinoid cell line), and this may not be fully representative of the behavior of real-life LNENs. Another limitation is the use of RAD001 as the only mTORi representative in the in vivo murine model, whereas the potential effect of Torin1 was not assessed in this setting—this decision being related to the toxicity that Torin1 showed in clinical context in other cancers. Further studies using more LNEN cell lines and animal models are therefore warranted before human studies should be designed; however, these results are encouraging, and suggest that lysosomal inhibitors such as CQ may be considered in the new therapeutic arsenal for patients with metastatic refractory LNENs, with or without mTORi.

## 5. Conclusions

The present study suggests that inhibition of autophagy by CQ robustly suppresses LNEN cell survival and proliferation, potentiating the anti-tumor effects of mTORi via stimulation of apoptosis. More research is therefore warranted to evaluate CQ efficacy alone or in combination with mTORi, and before considering its use in the clinical setting for patients with advanced progressing lung neuroendocrine neoplasms.

## Figures and Tables

**Figure 1 cancers-13-06327-f001:**
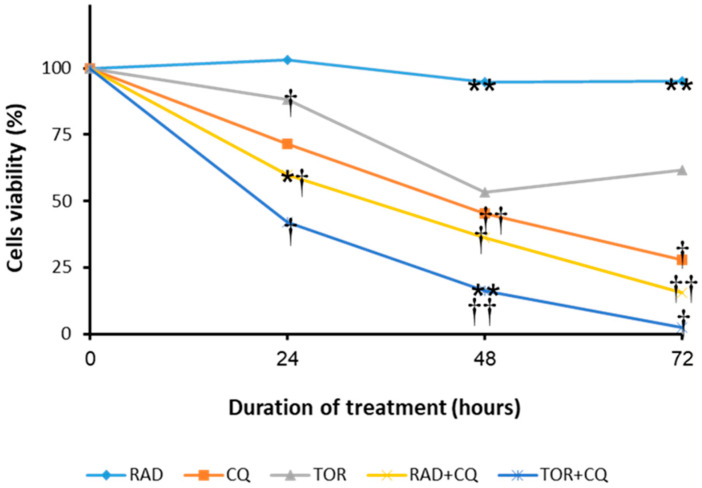
NCI-H727 cell viability after exposure to RAD001, Torin1, CQ, and their combinations (XTT method). The graph presents the relative reductions in cells’ viability for each treatment compared to DMSO-treated cells. Treatment with RAD and CQ, or with TOR and CQ, induced a significant decrease in cells’ viability compared to RAD alone at 24 h (*p* = 0.05); the effect was more pronounced after 48 h and 72 h. Compared to CQ alone, concomitant treatment with TOR and CQ showed a trend towards lower viability at 24 and 72 h (*p* = 0.06) and a significant decrease at 48 h (*p* = 0.01). RAD and CQ treatment showed a decrease in cells’ viability at 24 h (*p* = 0.05) and a trend towards lower viability at 48 and 72 h. † *p* < 0.05, †† *p* < 0.01, compared to treatment with RAD001. * *p* < 0.05, ** *p* < 0.01 compared to treatment with CQ. Abbreviations—RAD: RAD001; TOR: Torin1; CQ: chloroquine; values are given as the mean and SE of 5 independent experiments.

**Figure 2 cancers-13-06327-f002:**
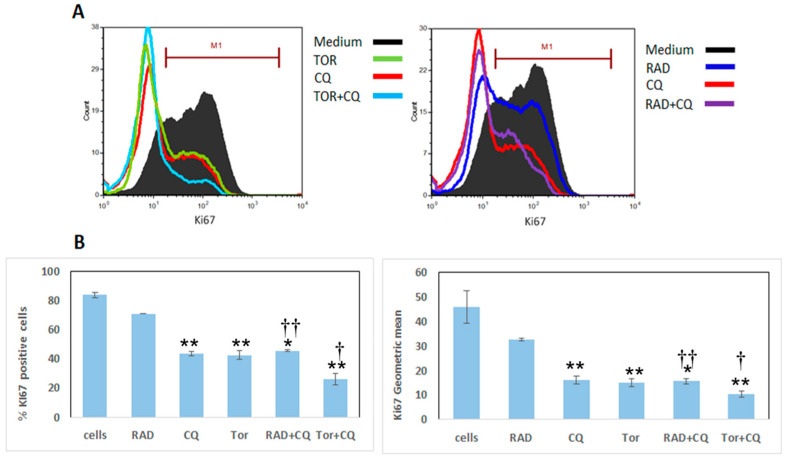
NCI-H727 cell proliferation after exposure to RAD001, Torin1, CQ, and their combinations. Panel (**A**) shows Ki-67-positive cell counts following 48 h of exposure to Torin1, CQ, and their combination (left) or RAD001, CQ, and their combination (right). Panel (**B**) shows the column bar presentation of panel (**A**). Left: the ratio of Ki-67-positive cells for each treatment. Treatment with CQ, Torin1, and CQ in combination with RAD001 or Torin1 showed decreased proliferation compared with cells without treatment (*p* = 0.002, 0.008, 0.01, and 0.008, respectively). The addition of CQ to RAD001 or to Torin1 was significantly more effective in decreasing proliferation (*p* = 0.008 and *p* = 0.04, respectively, in green) compared to RAD001 or Torin1 alone. * *p* < 0.05, ** *p* < 0.01, compared to untreated cells; † *p* < 0.05, †† *p* < 0.01, for comparison with and without CQ. Abbreviations—“Cells” refer to untreated cells; RAD: RAD001; TOR: Torin1; CQ: chloroquine. Values are given as the mean and SE of 3 independently performed experiments.

**Figure 3 cancers-13-06327-f003:**
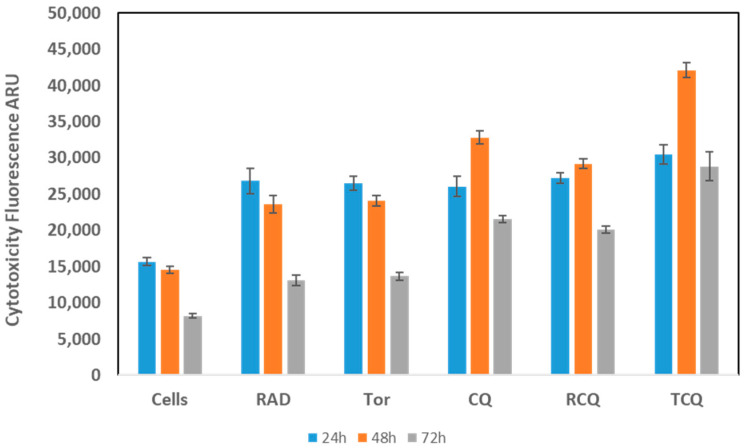
The effect of CQ and mTORi on NCI-H727 cell death. We followed the cells for 24, 48, and 72 h post-treatment. Interestingly, the effect was similar for all treatment arms at 24 h. In contrast, at 48 h, RAD001 and Torin1 showed a mild decrease in cytotoxicity, whereas CQ—alone, or in combination with RAD001 or Torin1—showed an increase in the cytotoxic effect; this pattern persisted at 72 h, showing a slight decrease in the CQ + RAD001 arm when compared with the 48 h effect. Abbreviations—“Cells” refer to untreated cells; RAD: RAD001; TOR: Torin1; CQ: chloroquine; RCQ: RAD001 + CQ; TCQ: Torin1 + CQ.

**Figure 4 cancers-13-06327-f004:**
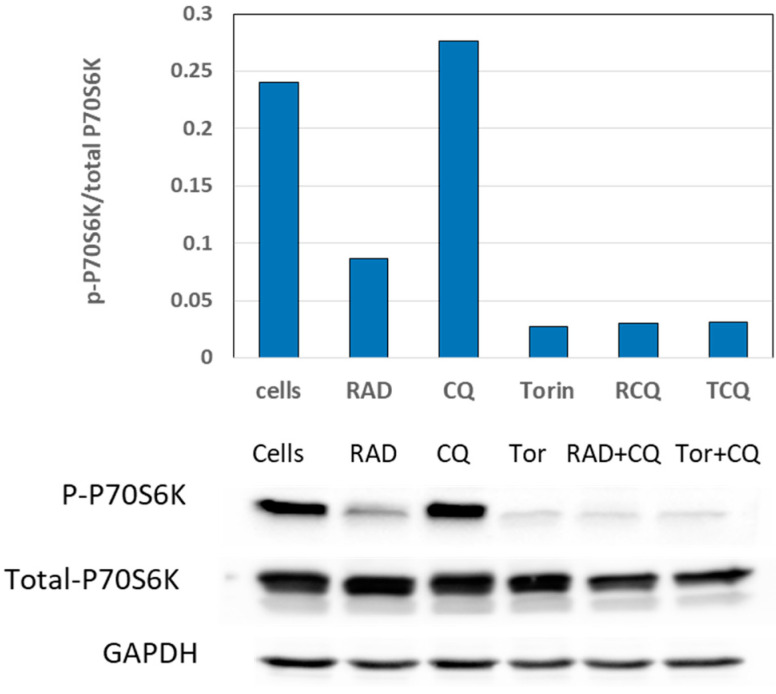
The effects of CQ and mTORi on the mTOR pathway. We evaluated the effects of RAD001, Torin1, CQ, and their combinations on p70-S6K phosphorylation by using Western blotting of total and phosphorylated p70-S6K and their quantification. GAPDH was used as a reference. As expected, both the mTORC1i RAD001 and the total mTORi Torin1 decreased the levels of p-P70S6K, whereas CQ alone increased the p-P70S6K phosphorylation. Importantly, the addition of RAD001 or Torin1 to CQ abolished the effect of the latter and decreased the p-P70S6K phosphorylation, diminishing the pro-proliferative activity induced by activation of mTOR. Abbreviations—“Cells” refer to untreated cells; RAD: RAD001; TOR: Torin1; CQ: chloroquine.

**Figure 5 cancers-13-06327-f005:**
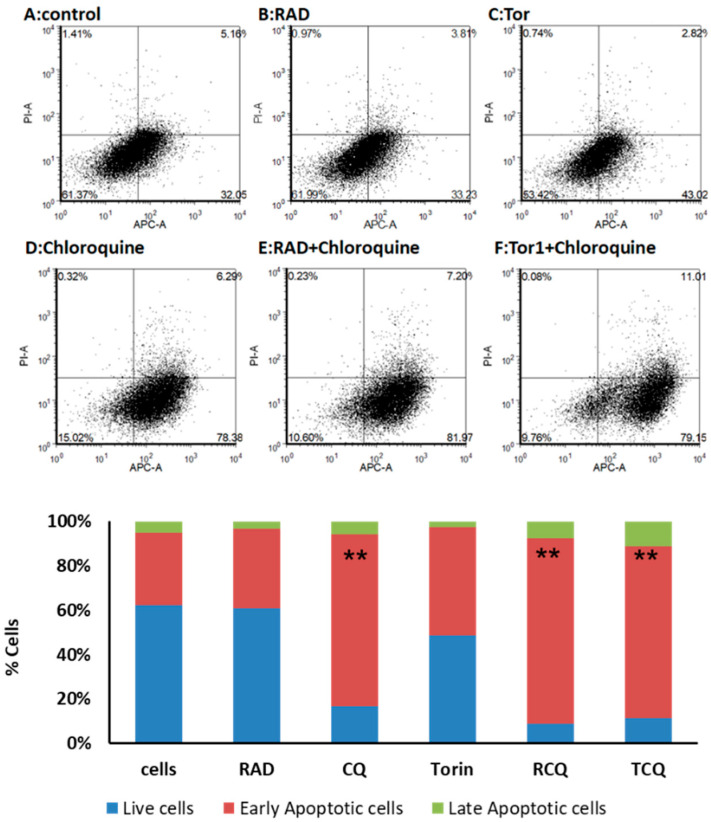
The effects of CQ and mTORi on apoptosis in NCI-H727 cells. We assessed Annexin V and PI as markers of the early stage of apoptosis via flow cytometry. We treated NCI-H727 cells with CQ, RAD001, Torin1, and their combinations (at the dosages mentioned in the Materials and Methods section), and looked at their effects on different stages of apoptosis (early, late, and total). At 48 h, CQ increased the number of apoptotic cells by 45%, whereas the effects of Torin1 and RAD001 were milder (a 17% and 3% increase, respectively) (**A**–**D**). The combination of CQ and RAD001 significantly increased the numbers of both early apoptotic cells (by 48%, *p* = 0.007) and total apoptotic cells (by 51%, *p* = 0.002) vs. RAD001 alone; the combination of CQ and Torin1 increased the number of apoptotic cells by 45%, inducing higher levels of late apoptotic cells (11%), but this was not statistically significant (**E**–**F**). ** *p* < 0.01 compared to treatment with RAD001. Abbreviations—“Cells” refer to untreated cells, as controls; RAD: RAD001; TOR: Torin1; CQ: chloroquine; RCQ: RAD001 + CQ; TCQ: Torin1 + CQ.

**Figure 6 cancers-13-06327-f006:**
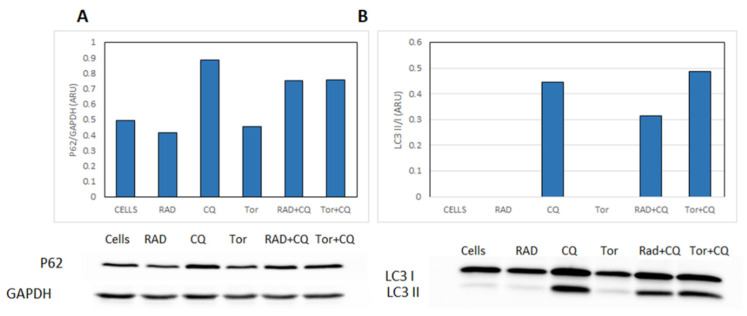
The effects of CQ and mTORi on the autophagy markers p62 and LC3-II in NCI-H727 cells. (**A**) Treatment with CQ increased the levels of p62, whereas RAD001 and Torin1 mildly decreased them, compared to untreated starved cells; the addition of CQ to either RAD001 or Torin1 increased p62 levels, supporting the effective suppression of autophagy. We did not find an increase in the autophagy flux following treatment with RAD001 or Torin1. (**B**) In untreated NCI-H727 cells, LC3-II was absent; CQ treatment increased LC3-II levels, indicating that autophagosome generation was intact. The addition of mTORi (RAD001 and Torin1) to CQ robustly increased LC3-II levels, suggesting that the mTORi increased the autophagic flux. “Cells” refer to untreated cells.

**Figure 7 cancers-13-06327-f007:**
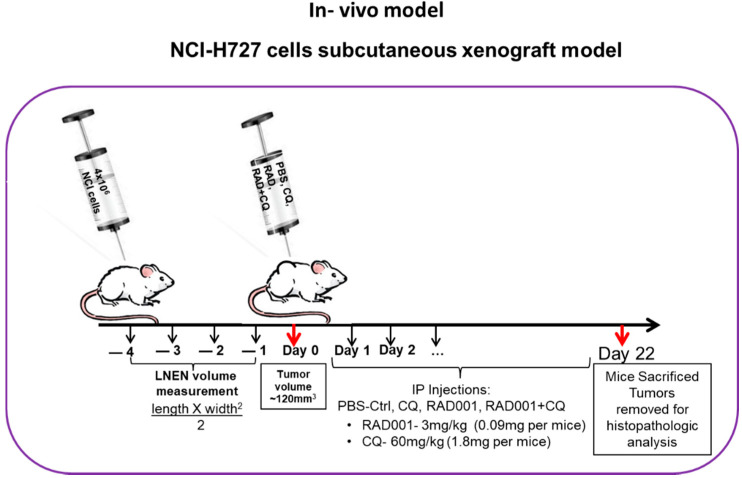
An illustration of the subcutaneous NCI-H727 murine xenograft model. Abbreviations—CQ: chloroquine; Ctrl: control; LNEN: lung neuroendocrine neoplasms; PBS: phosphate-buffered saline; RAD: RAD001.

**Figure 8 cancers-13-06327-f008:**
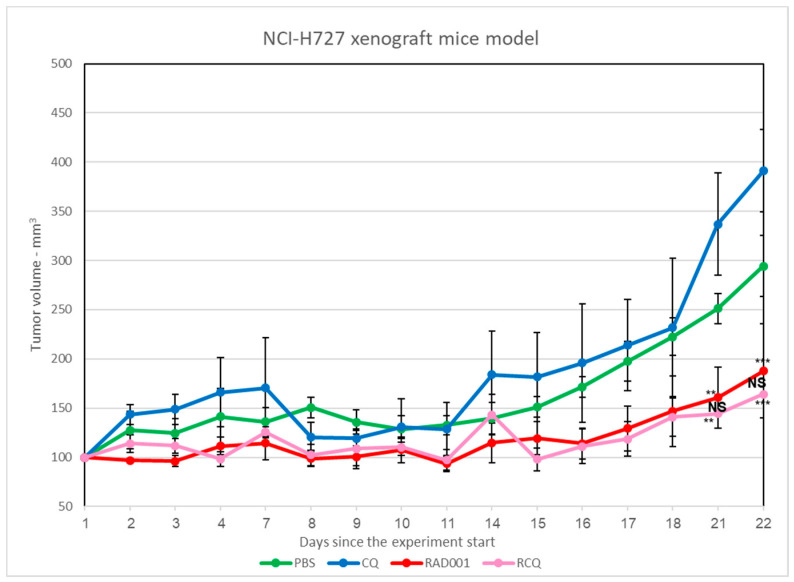
The effects of CQ and RAD001 in the NCI-H727 murine xenograft model. A total of 4 × 10^6^ NCI-H727 cells were subcutaneously injected into athymic nude mice, which were then randomized and treated daily with the following: (1) vehicle (PBS, 100 μL), (2) chloroquine (CQ) (60 mg/kg), (3) RAD001 (3 mg/kg), or (4) RAD001 and CQ (3 mg/kg and 60 mg/kg respectively), for the next 22 days. Tumor size was measured daily using a caliper, and tumor volume was calculated using the equation length× (width^2^)/2. Tumor volume measurements: the PBS-treated group (green line) developed large neoplasms; however, RAD001 (red line), as well as the combination of RAD001 and CQ (pink line), showed an inhibitory effect on tumor growth, with a decrease in tumor size. In the CQ group (blue line), the effect on the tumor size was less significant. PBS (*n* = 12), CQ (*n* = 10), RAD001 (*n* = 10), RAD001 and CQ (*n* = 11). Significant vs. CQ and PBS ** *p* < 0.01, *** *p* < 0.001; NS: not significant.

## Data Availability

The data presented in this study are available in this article (and Appendix A).

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
