# Peer review of "The Autophagy Inhibitor Chloroquine, Alone or in Combination with mTOR Inhibitors, Displays Anti-Tumor Effects in In Vitro and In Vivo Lung Carcinoid Models"

_cancers, 2021, doi:10.3390/cancers13246327_

Round 1

Reviewer 1 Report

This manuscript by Adi knigin et.al described the role of autophagy in neuroendocrine neoplasms of the lung (LNEN) and demonstrated autophagy inhibitor chloroquine (CQ), alone or combination with mTOR inhibitors could be a promising strategy for LNEN treatment. In this paper, CQ inhibitor was used alone or combine with mTOR inhibitors in vitro and in vivo to reduce tumor cell viability and growth by decreasing P62 level. This manuscript is coherent and quite well-written.

However, autophagy is a very important pathway and has different roles in tumor development, as a repressor or activator, so it is very important to know the mechanism behind in LNEN. For example, how about the effects of CQ and mTOR inhibitors in xenograft mice model by changing their administration time? The same time as NCI-H727 cell injection or one day, two days later?

Moreover, only one cell line NCI-H727 was used in vitro is not enough to make the conclusion. To make it solid, I think at least one or two more cell lines are needed.

Furthermore, there are no significant differences between the groups of RAD001 alone and combination with CQ in vivo. Which is very different with in vitro data, if so, how to demonstrate their validity?

Author Response

We thank the Reviewers for their helpful comments in an endeavor to improve the quality of this manuscript.

Reviewer#1

This manuscript by Adi Knigin et.al described the role of autophagy in neuroendocrine neoplasms of the lung (LNEN) and demonstrated autophagy inhibitor chloroquine (CQ), alone or combination with mTOR inhibitors could be a promising strategy for LNEN treatment. In this paper, CQ inhibitor was used alone or combine with mTOR inhibitors in vitro and in vivo to reduce tumor cell viability and growth by decreasing P62 level. This manuscript is coherent and quite well-written.

However, autophagy is a very important pathway and has different roles in tumor development, as a repressor or activator, so it is very important to know the mechanism behind in LNEN.

  1. For example, how about the effects of CQ and mTOR inhibitors in xenograft mice model by changing their administration time? The same time as NCI-H727 cell injection or one day, two days later?

We thank the reviewer for this suggestion. However, we would like to highlight that the model we used is a subcutaneous xenograft model; therefore, following the subcutaneous inoculation of the NCI-H727 cells, we had to wait for 4 to 6 days until the formation of the tumor (as previously published, Ref. 15), and only then to start the study drugs administration. The main aim of our study was to evaluate the effect of the drugs on tumor progression rather than their effect on tumor prevention.

  1. Moreover, only one cell line NCI-H727 was used in vitro is not enough to make the conclusion. To make it solid, I think at least one or two more cell lines are needed.

We thank the reviewer for this comments. We are well aware that the use of only one LNEN cell line represents one of the limitations of our study, and we have addressed this point already in the Discussion (revised manuscript lines 413-415). Noteworthy, we did considered and purchased another LNEN cell line for our study - the NCI-H720 cell line (CRL#5838; ATCC, Manassas, Virginia, USA) - unfortunately, we didn't succeed in growing the cells, after repeated attempts.

  1. Furthermore, there are no significant differences between the groups of RAD001 alone and combination with CQ in vivo. Which is very different with in vitro data, if so, how to demonstrate their validity?

We thank the reviewer for this observation. We are aware of the differences seen between the in vitro and in vivo models, and of the limitation of using cell lines models that are not always representative or imitating the in vivo models. Noteworthy, these results were shown on repeated experiments (at least 3 times), including a relative large number of mice (10 mice) for each in vivo experiment. We believe that future research will assist to decipher this different behavior and to validate the present results.     

Reviewer 2 Report

The present manuscript by Knigin A et al examines the effect of chloroquine alone or in combination to mTOR inhibitors on in-vitro and in-vivo models of lung NENs. The authors conclude that

Minor comments

The abbreviation mTORi should be written earlier thus in line 13 and not in line 18.

Major comments

The bronchial carcinoid cell line NCI-H727 is considered to be a typical lung carcinoid.  There are also cell lines available eg NCI-H720 corresponding to atypical lung carcinoid. It is not explained why the authors did not include an atypical lung carcinoid cell line. Otherwise, the authors have to clarify that their results concern only typical lung carcinoids.

At the discussion when the in vivo results were commended the authors do not provide an explanation why the CQ alone didn´t show a decrease in the neoplasm size but that the in vivo model did develop smaller tumors when CQ was combined with RAD001 (comparable to the effect of RAD001 alone).

Author Response

Reviewer#2

The present manuscript by Knigin A et al examines the effect of chloroquine alone or in combination to mTOR inhibitors on in-vitro and in-vivo models of lung NENs. The authors conclude that…

Minor comments

  1. The abbreviation mTORi should be written earlier thus in line 13 and not in line 18.

We thank the reviewer for this suggestion and corrected the abbreviation for mTORi in the revised manuscript as required (Revised manuscript, Abstract, highlighted, page 2).

Major comments

  1. The bronchial carcinoid cell line NCI-H727 is considered to be a typical lung carcinoid.  There are also cell lines available eg NCI-H720 corresponding to atypical lung carcinoid. It is not explained why the authors did not include an atypical lung carcinoid cell line. Otherwise, the authors have to clarify that their results concern only typical lung carcinoids.

We thank the reviewer for this comment, which was already addressed in the response to Reviewer #1 comment #2. As mentioned, we are well aware that the use of only one LNEN cell line represents one of the limitations of our study, and we have addressed this point already in the Discussion (Revised manuscript, Discussion, highlighted, page14). Noteworthy, we purchased another LNEN cell line for our study - the NCI-H720 atypical lung carcinoid cell line (CRL#5838; ATCC, Manassas, Virginia, USA) – unfortunately, we didn't succeed in growing the cells, after repeated attempts.  

  1. At the discussion when the in vivo results were commended the authors do not provide an explanation why the CQ alone didn´t show a decrease in the neoplasm size but that the in vivo model did develop smaller tumors when CQ was combined with RAD001 (comparable to the effect of RAD001 alone).

We thank the reviewer for this observation. Indeed, and unexpectedly, the tumor xenografts were less responsive to CQ alone, in contrast with both the in-vitro cell-line results as well as with our previous studies in a pancreatic neuroendocrine tumor model (Ref. 15). Although the reason for this different behavior remains unclear, we believe that it is possibly related to physiological differences between cell-lines and xenografts that impact drug absorption, distribution, metabolism and excretion, and drug-to-drug interaction. Moreover, we (Ref.11, Ref. 15) and others (25) have shown that CQ can be additive to RAD001 in reducing tumor cell viability in parallel with increasing apoptosis and reducing autophagy. We added an explanatory sentence in the revised manuscript (Revised manuscript, Discussion, highlighted, page 13).

Reviewer 3 Report

Article

The autophagy inhibitor chloroquine, alone or in combination with mTOR inhibitors, induces a dichotomous lethal effect in lung neuroendocrine tumor in-vitro and in-vivo models.

Adi Knigin, Shani Avniel-Polak, Gil Leibowitz, Kira Oleinikov, David J. Gross and Simona Grozinsky-Glasberg.

In this article Knigin and coworkers examined the effect of chloroquine and mTORi on lung neuroendocrine neoplasm (LNEN) models. To achieve this purpose, they performed in vitro experiments using a LNEN cell line (NCI-H727) treated with chloroquine (CQ) +/- mTORi RAD001 and Torin1 and examined cell viability, proliferation and cytotoxicity. They also performed in vivo experiments establishing a xenograft model and analyzing the effect of tested drugs on tumor growth. The main result of this project is that CQ is able to suppress NCI-H727 viability and proliferation and to increase cytotoxicity and apoptosis and the co-presence of an mTORi exacerbates these effects. The mechanism trough which CQ and mTORi would affect LNEN viability involves p62 and LC3.

The paper although being rich in some quality data, is poorly organized and written and thus cannot convince the readership that it is a sequential and logical study. I have no option but to reject this paper in its present form. Here my comments:

Title is misleading and the dichotomous effect of CQ is unclear and should be empathized; I suggest to streamline the title

Results are merely descriptive and concepts do not provide a causal link; it is unclear which logical passages authors did. Moreover, conclusions should be included at the end of each result section to summarize the highlights

There are many spelling mistakes and the manuscript needs to be carefully edited for English language. The authors should seek editorial assistance to improve this

The discussion is superficial and doesn't study in deep nor hypothesize the impact of research on clinical practice.

Please check and implement bibliography since many assumptions are not properly justified by references (e.g lines 74-76, line 151, lines 287-288; line 417)

References 14 and 23 are referring to the same work. Please check and correct

I suggest to move “Reagents” at the beginning of Materials & Methods section

Primary antibodies mentioned in reagents section should be moved in the opportune section. Are they have been used for western Blotting?

Informations on reagent suppliers are often missing and should be properly included in Material and Methods section.

The material and methods section is rudimentary since many informations are missing (e.g NCI-H727 provider in 2-1; XTT assay reader used in 2.2, cell density used in 2.4, image analysis software used in 2.5, secondary antibodies used in 2.5)

Drugs concentration and treatment lengths should be properly justified according to literature (see PMID: 19150980; PMID: 21576371, PMID: 25654547)

Results title should not contain (and start with) Figures. Please move all figures at the end or betwixt each result section.

Lines 152-153: what about mentioned histopathological analysis?

Figure 7 corresponds (with some slight differences) to Figure 1 (S Avniel-Polal et al., Endocrine Related Cancer 2018). A little “graphical effort” would be very appreciated.

In Figure 1, 24h time point differences do not reach any statistical significance? Please check

Between lines 175 and 176 there is a double spacing. Please check

Figures contains “cells” sample. What exactly does it refer to?

What is the meaning of “Ki67 geometric mean”? What can it bring to the data? It seems to be redundant.

Figure 3: do authors have any explanation for the difference observed at 72h between the two assays? Is the % of permeable membrane an alternative readout of cell death and therefore a different way to measure drugs cytotoxicity?

Figure 5: do authors have any explanation for the 40% of cell death in control cells?

Figure 8: I suggest to remove “day” from each time point leaving only numbers

To better appreciate autophagosomal formation and to prevent lysosomal degradation of autophagosome associated LC3b, it will be useful to use a lysosomial acidification inhibitor (e.g Bafilomycin A1).

Author Response

We thank the Reviewers for their helpful comments in an endeavor to improve the quality of this manuscript.

Reviewer#3

In this article Knigin and coworkers examined the effect of chloroquine and mTORi on lung neuroendocrine neoplasm (LNEN) models. To achieve this purpose, they performed in vitro experiments using a LNEN cell line (NCI-H727) treated with chloroquine (CQ) +/- mTORi RAD001 and Torin1 and examined cell viability, proliferation and cytotoxicity. They also performed in vivo experiments establishing a xenograft model and analyzing the effect of tested drugs on tumor growth. The main result of this project is that CQ is able to suppress NCI-H727 viability and proliferation and to increase cytotoxicity and apoptosis and the co-presence of an mTORi exacerbates these effects. The mechanism trough which CQ and mTORi would affect LNEN viability involves p62 and LC3.

The paper although being rich in some quality data, is poorly organized and written and thus cannot convince the readership that it is a sequential and logical study. I have no option but to reject this paper in its present form. Here my comments:

  1. Title is misleading and the dichotomous effect of CQ is unclear and should be empathized; I suggest to streamline the title.

We thank the reviewer for this suggestion, and we changed the title, as recommended, to a more focused one (Revised manuscript, Title, highlighted, page 1).

  1. Results are merely descriptive and concepts do not provide a causal link; it is unclear which logical passages authors did. Moreover, conclusions should be included at the end of each result section to summarize the highlights

We thank the reviewer for the criticism. We have addressed these comments point-by-point and re-write the results in a more causal-effect and clearer way; moreover, we added specific conclusions at the end of each result and highlighted the rationale behind the specific experiment (Highlighted, Revised Manuscript, Results).

  1. There are many spelling mistakes and the manuscript needs to be carefully edited for English language. The authors should seek editorial assistance to improve this.

We thank the reviewer for this comment. The manuscript underwent a throughout English language editing, as requested.

  1. The discussion is superficial and doesn't study in deep nor hypothesize the impact of research on clinical practice.

We thank the reviewer for the criticism. The present study suggests that inhibition of autophagy by CQ robustly inhibits LNEN cell survival and proliferation, potentiating the anti-tumoral effects of mTORi via stimulation of apoptosis. This may have important clinical implications for considering the use of CQ as a therapeutic strategy aiming to overcome mTORi-related drug resistance and represents an important consideration for the development of clinical trials in patients with metastatic or refractory LNEN. We have revised the discussion and rewrite parts of it, addressing the different stages of the study as well as the possible impact on the clinical practice (Revised manuscript, Discussion, pages 11-14, highlighted).

  1. Please check and implement bibliography since many assumptions are not properly justified by references (e.g lines 74-76, line 151, lines 287-288; line 417)

We thank the reviewer for this comment. We added adequate references, as required (Revised manuscript, page 6, Highlighted)

  1. References 14 and 23 are referring to the same work. Please check and correct

We corrected the references and deleted the repetition.

  1. I suggest to move “Reagents” at the beginning of Materials & Methods section.

We moved the "Reagent" section at the beginning of Material and Methods, as suggested (Revised manuscript, Reagents, page 4, highlighted).

  1. Primary antibodies mentioned in reagents section should be moved in the opportune section. Are they have been used for western Blotting?

We moved the antibodies from "Reagents" to the specific related section entitled "Western Blot", as suggested (Revised manuscript, Western Blot, page 6, highlighted).

  1. Informations on reagent suppliers are often missing and should be properly included in Material and Methods section.

We thank the reviewer for this comment and provided the requested information on the different providers (Revised manuscript, Material and Methods, Pages 4-6, highlighted).

  1. The material and methods section is rudimentary since many informations are missing (e.g NCI-H727 provider in 2-1; XTT assay reader used in 2.2, cell density used in 2.4, image analysis software used in 2.5, secondary antibodies used in 2.5)

We provided the specific missing information (Revised manuscript: Cell culture, NCI-H727 provider, page 4; Cell viability, XTT assay reader, page 5; Cell viability, specific cell density used, page 5; Flow cytometry, specific antibody, page 5 – all highlighted).

  1. Drugs concentration and treatment lengths should be properly justified according to literature (see PMID: 19150980; PMID: 21576371, PMID: 25654547)

We thank the reviewer for this suggestion and for referral to the specific publications. As specified in the manuscript, we decided on which dosages and times to use based on the IC50 of CQ and Torin1 at 48 hours. As mentioned in the text, the IC50 for RAD001 in NCI-H727 cells could not be determined because RAD001 did not show a significant inhibitory effect on these cells in the preliminary IC50 experiments. Therefore, we decided to use the RAD001 IC50 from our previous experience in another neuroendocrine tumor cell line, specifically, the BON-1 cell line (Revised manuscript, Cell viability, page 5, highlighted, and Ref. 11). Noteworthy, there was no problem to induce autophagy with the dosages we used for the mTORi (as shown by the levels of p62 and LC3-II on Western blot; Revised manuscript, Figure 6).

  1. Results title should not contain (and start with) Figures. Please move all figures at the end or betwixt each result section.

We completely agree with this comment. However, the location of the Figures at the beginning of the Results was not related or depending on our original manuscript, but rather to its automatic "transformation" by the journal template itself, following submission. It should disappear at the final editing.

  1. Lines 152-153: what about mentioned histopathological analysis?

We thank the reviewer for this comment. At the completion of the experiments mice were sacrificed, the tumors were excised and weighted, followed by histopathological analysis including H&E staining and validation of a neuroendocrine tumor morphology. We added this explanation to the manuscript (Revised manuscript, "Subcutaneous xenograft mouse model", page 6-7, highlighted).

  1. Figure 7 corresponds (with some slight differences) to Figure 1 (S Avniel-Polak et al., Endocrine Related Cancer 2018). A little “graphical effort” would be very appreciated.

We thank the reviewer for this comment. We indeed used the same platform for the development of the NCI-H727 subcutaneous xenografts mice model as we did in our previous research with pancreatic NEN mice models; however, both the platform as well as the referred figure, were created by ourselves. We addressed this issue now in the revision (Revised manuscript, Figure 7 - adapted, with graphical changes as appropriate, from our previous publication; page 7, highlighted) (Ref. 15).

  1. In Figure 1, 24h time point differences do not reach any statistical significance? Please check

We thank the reviewer for this observation and corrected the figure including statistical significance at 24h time point, as appropriate (Revised manuscript, Figure 1).

  1. Between lines 175 and 176 there is a double spacing. Please check

This happened unrelated to our original submission, again, due to the automatic uploading of the manuscript into the journal's template. It should disappear at the final editing.

  1. Figures contains “cells” sample. What exactly does it refer to?

We thank the reviewer for this comment. The treatment group named “cells” in the figures refer to untreated cells (as the experiment control). We have corrected and explained it in the figures legends. (Revised manuscript, Figures 2, 3, 4, 5, 6, and the specific Figure legends).

  1. What is the meaning of “Ki67 geometric mean”? What can it bring to the data? It seems to be redundant.

We thank the reviewer for the comment. We believe that the two Ki67 analyses (Figure 2A and 2B) are complementary. The positive Ki67 cells signify the amount of cells undergoing proliferation, while the geometric mean represented the level of proliferation. Taken together, the amount of proliferating cells (as Ki67 positive cells) and the proliferating levels of the cells (as Ki67 geometric mean) are two parts of the same story, and in our view are important to understand the results.

  1. Figure 3: do authors have any explanation for the difference observed at 72h between the two assays? Is the % of permeable membrane an alternative readout of cell death and therefore a different way to measure drugs cytotoxicity?

This represent the same data but with a different analysis, as in the lower panel we normalized each treatment to the control at the same time point. It helps to see more clearly the effect of CQ on cell cytotoxicity. However, to prevent redundancy, we decided to keep only one figure (Revised manuscript, Figure 3).

  1. Figure 5: do authors have any explanation for the 40% of cell death in control cells?

We thank the reviewer for this important comment. As with any cell line, some cells undergo death as part of their growth. While 40% is high, it may reflect the normal proliferation/death equilibrium of NCI-H727 cells. Nevertheless, the effect of the treatment significantly surpasses the baseline viability reads of the control, thus proving that the treatment indeed reduced cell viability. 

  1. Figure 8: I suggest to remove “day” from each time point leaving only numbers

We thank the reviewer for the suggestion and removed the word "day" from each time point in Figure 8.

  1. To better appreciate autophagosomal formation and to prevent lysosomal degradation of autophagosome associated LC3b, it will be useful to use a lysosomal acidification inhibitor (e.g Bafilomycin A1).

We thank the reviewer for this idea. However, according to the literature, both compounds (CQ and bafilomycin, BafA1) impair autophagosomes and lysosomes fusion. CQ inhibition of autophagy is mediated through blocking autophagosome-lysosome fusion and not via the degradation capacity of lysosomes as previously assumed. BafA1, in contrast, inhibits the degradation capacity of lysosomes by decreasing their acidity, but it can also impair fusion between autophagosomes and lysosomes (see reference PMID: 29940786). Moreover, CQ and HCQ (hydroxychloroquine) are the only FDA-approved drugs inhibiting autophagy used in clinical practice, therefore we chose to continue to work with CQ. We have previously worked with BafA1 as a control to our experiments, which gave similar effect to CQ treatment in our cell model.

Round 2

Reviewer 3 Report

All suggestions and comments have been satisfactorily examined. I therefore believe that the work can be published in the present form.